# Catalyst Speciation during *ansa*-Zirconocene-Catalyzed Polymerization of 1-Hexene Studied by UV-vis Spectroscopy—Formation and Partial Re-Activation of Zr-Allyl Intermediates [note 1]

**DOI:** 10.3390/polym11060936

**Published:** 2019-05-29

**Authors:** Valentina N. Panchenko, Dmitrii E. Babushkin, John E. Bercaw, Hans H. Brintzinger

**Affiliations:** 1Boreskov Institute of Catalysis, Russian Academy of Sciences, Siberian Branch, RU-630090 Novosibirsk, Russian; panchenko@catalysis.ru (V.N.P.); dimi@catalysis.ru (D.E.B.); 2Novosibirsk State University, Pirogova Str. 2, 630090 Novosibirsk, Russian; 3Arnold and Mabel Beckman Laboratories of Chemical Synthesis, California Institute of Technology, Pasadena, CA 91125, USA; 4Fachbereich Chemie, Universität Konstanz, D-78464 Konstanz, Germany; hans.brintzinger@uni-konstanz.de

**Keywords:** polymerization catalysis, UV-vis spectroscopy, catalyst speciation, zirconocene-allyl cations, de-activation, re-activation

## Abstract

Catalyst speciation during polymerization of 1-hexene in benzene or toluene solutions of the catalyst precursor SBIZr(μ-Me)_2_AlMe_2_^+^ B(C_6_F_5_)_4_^−^ (SBI = *rac*-dimethylsilyl-bis(1-indenyl)) at 23 °C is studied by following the accompanying UV-vis-spectral changes. These indicate that the onset of polymerization catalysis is associated with the concurrent formation of two distinct zirconocene species. One of these is proposed to consist of SBIZr-σ-polyhexenyl cations arising from SBIZr-Me^+^ (formed from SBIZr(μ-Me)_2_AlMe_2_^+^ by release of AlMe_3_) by repeated olefin insertions, while the other one is proposed to consist of SBIZr-η^3^-allyl cations of composition SBIZr-η^3^-(1-R-C_3_H_4_)^+^ (R = n-propyl), formed by σ-bond metathesis between SBIZr-Me^+^ and 1-hexene under release of methane. At later reaction stages, all zirconocene-σ–polymeryl cations appear to decay to yet another SBIZr-allyl species, i.e., to cations of the type SBIZr-η^3^-(x-R-(3-x)-pol-C_3_H_3_)^+^ (pol = *i*-polyhexenyl, x = 1 or 2). Renewed addition of excess 1-hexene is proposed to convert these sterically encumbered Zr-allyl cations back to catalytically active SBIZr-σ–polymeryl cations within a few seconds, presumably by initial 1-hexene insertion into the η^1^- isomer, followed by repeated additional insertions, while the initially formed, less crowded allyl cations, SBIZr-η^3^-(1-R-C_3_H_4_)^+^ appear to remain unchanged. Implications of these results with regard to the kinetics of zirconocene-catalyzed olefin polymerization are discussed.

## 1. Introduction

In recent years, zirconocene-based catalyst systems for the polymerization of olefins have found widespread application for the production of polyolefin materials with special properties [1]. Concomitantly, considerable research efforts have been devoted to the identification and quantification of zirconocene complex species which arise during polymerization catalysis and to their effects on the outcome of the catalytic process [2,3,4]. Of particular interest in this regard is the question of which zirconocene species constitute the main resting state, i.e., the predominant complex species, during ongoing catalysis. 

Possible resting states that have been considered to date include Zr-polymeryl cations carrying σ-bound, regio-regular (1,2-inserted) [5,6] or regio-irregular (2,1-inserted) chain ends [7], either in inner-sphere association with a counter-anion [6], in agostic contact with the Zr center [8], or as an AlMe_3_ adduct [9]. At lower temperatures, under conditions of ‘living’ polymerization, Zr-σ-polymeryl cations with regioregular Zr-bound chain ends have been conclusively identified and thoroughly studied with respect to their reactivity toward olefin insertion through ^1^H NMR spectroscopy by Landis and coworkers [5,6]. At ambient temperature, σ-bound Zr-polymeryl cations appear to be too short-lived to be observable by standard ^1^H or ^13^C NMR techniques. Hilty and coworkers have recently shown, however, that even at ambient temperature cationic Zr-σ-polymeryl species can be observed and followed in their olefin-insertion kinetics by means of ^13^C NMR spectroscopy using hyperpolarized 1-hexene [10]. The role of Zr-σ-polymeryl cations in zirconocene-catalyzed olefin polymerization thus appears to be clearly established by now. 

To which degree other zirconocene complexes likewise contribute to speciation and reaction kinetics in such catalyst systems still appears not entirely clear, however. Zirconocene cations with Zr-bound allylic chain ends, in particular, have also been implicated, already early on, as constituents of such catalyst systems [11,12,13,14]. Formation of zirconocene-η^3^-allyl cations under various reaction conditions, their structural fluxionality and possible pathways by which these species might intervene in zirconocene-catalyzed olefin polymerization have been extensively studied [15,16,17,18,19,20,21,22,23,24,25,26,27]. These Zr-allyl species are generally considered to be catalyst deactivation products, however, rather than active ingredients of such catalyst systems. 

Recently, some of us have shown that the start of olefin polymerization by various zirconocene-based catalyst systems is accompanied by the appearance of a UV-vis band, which resembles absorbance bands of *bona-fide* zirconocene-η^3^-allyl cations with respect to its absorbance maximum at λ_max_ = 560 nm and its broad band shape [28,29]. This has led to the proposal that Zr-η^3^-allyl cations arise in such catalyst systems already at an early stage of polymerization catalysis, rather than as “late” deactivation products. 

In order to clarify how such Zr-allyl cations are involved in zirconocene-catalyzed olefin polymerizations, we have studied the accompanying UV-vis spectral changes in more detail. For this study we chose again the ion pair SBIZr(μ-Me)_2_AlMe_2_^+^ B(C_6_F_5_)_4_^−^ (SBI = *rac*-dimethylsilyl-bis(1-indenyl)) as pre-catalyst, since cationic AlMe_3_ adducts of this type, together with complex anions derived from methylalumoxane (MAO) [30], represent the main pre-catalyst species in practically useful zirconocene-based polymerization catalysts activated by MAO [31,32,33,34,35,36]. 

## 2. Materials and Methods 

Air-sensitive compounds and solutions were handled in an argon-filled glove-box or in vacuo by means of break-seal techniques. Glassware was dried by heating overnight to 110 °C. Hydrocarbon solvents (benzene, toluene) were dried by treatment with freshly cut sodium at 50 °C for several days and distilled in vacuo, and subsequently degassed and stored under argon in a glovebox. 

Zirconocene complexes, obtained from MCAT Co., Konstanz, trityl perfluorotetraphenyl borate and trimethylaluminum (TMA), obtained from Aldrich Chemical Co., were used without further purification. CAUTION: TMA is pyrophoric and must be handled under thorough exclusion of air.

Solutions of the pre-catalyst SBIZr(μ-Me)_2_AlMe_2_^+^ B(C_6_F_5_)_4_^−^ were prepared in the following manner: To a solution containing 10 μmol of trimethylaluminum (TMA) in 10 mL of benzene was first added 2.05 mg (5 μmol) of SBIZrMe_2_ and then, after dissolution of the latter, 4.6 mg (5 μmol) of trityl perfluorotetraphenyl borate. The orange-red solution thus obtained contained the ion pair SBIZr(μ-Me)_2_AlMe_2_^+^ (F_5_C_6_)_4_B^−^ in a concentration of ca. 0.5 mmol/L and excess TMA in a nominal ratio of [Al]/[Zr] = 1, some of which is presumably consumed, however, by reaction with trace impurities. 

The orange-red solutions of the ion pair SBIZr(μ-Me)_2_AlMe_2_^+^ (F_5_C_6_)_4_B^−^ thus prepared showed ^1^H NMR spectra signals due to the cation SBIZr(μ-Me)_2_AlMe_2_^+^ [31], excess TMA, one equivalent of triphenylethane as well as of traces of methane and of the binuclear methylene-bridged cation (SBIZr)_2_(μ-CH_3_)(μ-CH_2_)^+^ [28]. The concentration of the latter remained unchanged during the subsequent catalysis reaction.

For the measurement of UV-vis spectra, a 2.5-mL sample of this pre-catalyst solution was placed, inside a glovebox, into a 1-cm UV-vis cuvette equipped with a teflon-lined septum screw cap. After transferring the cuvette to a Cary-60 spectrometer and measuring initial spectra, polymerization was started by injecting a volume of 0.1 mL (0.8 mmol) of 1-hexene through the teflon-lined septum closure, so as to produce an initial 1-hexene concentration of [hex] = 0.32 M, i.e., a ratio of [hex]/[Zr] = 640. Spectra were obtained using the kinetics software of the spectrometer with a scan rate of 24,000 nm per min, set at a cycle time of 0.2 min for the first two minutes of each run and at cycle times of 0.5 and 1 min after reaction times of two and of ten minutes, respectively. After a reaction time of 30 min, a second portion of 0.1 mL of 1-hexene was injected into the cuvette and UV-vis spectra were measured for another 30 min.

To measure, in parallel to these UV-vis spectra, also ^1^H NMR spectra of the same catalyst system, a 3-mL sample of the pre-catalyst solution described above was placed, inside a glovebox, into an NMR tube with 10 mm outer diameter, equipped with a septum closure. After transfer to the cavity of a Bruker AV400 spectrometer (Bruker BioSpin GmbH, Rheinstetten, Germany), initial NMR spectra were obtained at 400 MHz, using 90° pulses (24 ms) with 2.7 s acquisition time and 1 s delay time, while suppressing the signals of the benzene solvent by an appropriate pulse program. Polymerization was then started by injecting a volume of 0.15 mL (1.2 mmol) of 1-hexene through the septum closure of the NMR tube, so as to produce an initial 1-hexene concentration of [hex] = 0.4 M, with a ratio of [hex]/[Zr] = 800. Subsequent ^1^H NMR spectra were recorded approximately every 50 s. For each spectrum, 4 scans (plus two dummy scans) were measured with 5 s delay time and 2.8 s acquisition time.

UV-vis and ^1^H NMR spectra in 5 mm OD NMR tubes were obtained on Cary 60 (Agilent Technologies, Mulgrave, Victoria, Australia), Cary 500 (Agilent Technologies, Santa Clara, CA, USA), Bruker AV400 and Varian 500 MHz spectrometers (Agilent Technologies, Santa Clara, CA, USA).

## 3. Results

### 3.1. Qualitative UV-vis and ^1^H NMR Observations

When 1-hexene is added to a solution of SBIZr(μ-Me)_2_AlMe_2_^+^ (F_5_C_6_)_4_B^−^, prepared as described above, the reaction mixture changes color from orange-red to violet-blue, due to a decline of the absorbance of the cation SBIZr(μ-Me)_2_AlMe_2_^+^ at 495 nm and a concomitant rise of a new absorbance band at 560 nm. Both changes occur rapidly initially (Figure 1, stage 1), slow down after about two minutes and come to a standstill after about ten minutes (Figure 1, stage 2). Thereafter, absorbance values at both wavelengths undergo a slow reversal, indicating a partial re-conversion of the violet-colored catalyst species to the orange-red pre-catalyst, presumably by chain transfer of SBIZr-polyhexyenyl^+^ to Al_2_Me_6_ and complexation of SBIZrMe^+^ with AlMe_3_ after depletion of 1-hexene (Figure 1, stage 3). 

At first sight, these smooth, mutually complementary spectral changes would indicate an uncomplicated, partly reversible conversion of SBIZr(μ-Me)_2_AlMe_2_^+^ to some other zirconocene complex. Closer inspection of the UV-vis spectra reveals, however, that at least two distinct zirconocene species are formed from the pre-catalyst during each experiment. During a first stage of the reaction, which lasts for about 120 s, consecutive spectra of the reaction mixture show a sharp isosbestic point at 532 nm (Figure 1, stage 1). Prima facie, this would indicate the presence of only two species in the reaction system during that stage, i.e., a clean transformation of the pre-catalyst complex (to be labelled **C-0**) to a first conversion product, to be labeled **C-1**.

In the course of a second stage, however, at reaction times of ca. 120 to 600 s, the isosbestic point at 532 nm vanishes, as the absorbance at that wavelength increases significantly (Figure 1 D). This observation indicates that at least three species are being interconverted now. A further spectrally distinguishable product, to be labelled **C-2**, thus appears to arise in the reaction system, together with **C-0** and **C-1**, during stage 2. At still longer reaction times, the absorbance at 560 nm slowly declines and that at 495 nm is slowly restored, obviously by partial regeneration of the pre-catalyst SBIZr(μ-Me)_2_AlMe_2_^+^. A new isosbestic point is now established at 529 nm (Figure 1, stage 3). This indicates that only two species remain in the reaction system during stage 3, namely the more long-lived conversion products **C-2** and the slowly regenerating pre-catalyst **C-0,** while species **C-1** has apparently vanished by now.

^1^H NMR measurements conducted in parallel to the UV-vis measurements provide useful information regarding the concentration changes of the pre-catalyst species **C-0**. By following the intensity of its sharp signals at −0.65 and −1.39 ppm, due to its terminal AlMe_2_ and bridging Zr(μ-Me)_2_Al groups, its concentration can be estimated to decline to ca. 3–5% of its initial value in the course of ca. 100 s and to grow again to 10–15% of its initial value at longer reaction times. ^1^H NMR signals with a time course similar to that of the intermediary species **C-1** were not detectable during polymerization catalysis. At longer reaction times, weak ^1^H NMR signals arise at ca. 4.5–6.5 ppm (see Appendix A), in a region where signals of cationic Zr-allyl species have been observed [24,25,26]. These signals are too non-distinct, however, for any structural assignment. 

### 3.2. UV-vis Spectra of SBIZr-Alkyl Cations

The appearance and eventual decay of chain-propagating SBIZr-polymeryl^+^ cations during polymerization catalysis is expected to contribute to the UV-vis spectral changes described above. We have thus sought to obtain UV-vis spectra of stable SBIZr-alkyl ion pairs, such as SBIZr-Me^+^ B(C_6_F_5_)_4_^−^, to be used as a reference to recognize the spectral features of SBIZr-polymeryl^+^ cations in our catalyst systems. 

While solutions of the ion pair SBIZr-Me^+^ B(C_6_F_5_)_4_^−^ had been prepared and characterized by NMR spectroscopy in the presence of excess trityl perfluorotetraphenyl borate by Bochmann and coworkers [37], it proved difficult to generate solutions containing the cation SBIZr-Me^+^ in the absence of trityl cation. Reaction of SBIZrMe_2_ with one equivalent of trityl perfluorotetraphenyl borate proceeds via binuclear intermediates, (SBIZr-Me)_2_(μ-Me)^+^ [31], which react very sluggishly with the decreasing remains of the trityl cation. Any excess of trityl cation, on the other hand, causes severe spectral interference due to its very intense absorbance band centered at 425 nm. 

In a typical experiment, a solution of 1.95 mg (2 μmol) of trityl perfluorotetraphenyl borate in 1.5 mL of toluene-d_8_ was added dropwise to a solution of 0.83 mg (2 μmol) of SBIZrMe_2_ in 0.5 mL of toluene-d_8_. ^1^H NMR and UV-vis spectra of the resulting reaction mixture were both measured in a 5-mm OD NMR tube. ^1^H NMR spectra showed mainly the signal of the SBIZr-Me^+^ cation at −0.78 ppm [37], together with small signals of the diastereomeric binuclear intermediates, at −0.99 and −2.42 ppm (major diastereomer) and at −1.03 and −3.11 ppm (minor diastereomer) [31]. These signals decreased with time and vanished after ca. 4 h, thus leaving only the signals of the cation SBIZr-Me^+^. The UV-vis absorbance of the trityl cation at 425 nm likewise decreased at first, but finally remained constant at a value of ca 1.60, indicating that an excess of ca. 0.1 μmol of trityl cation remained in the reaction mixture. 

Addition of a small amount of SBIZrMe_2_ (ca. 0.1 mg) eliminated this UV-vis absorbance of the trityl cation completely. The reaction mixture then showed a UV-vis absorbance band with a maximum at 485 nm. Closely similar spectra are obtained when appropriate proportions of the absorbance of the trityl cation are subtracted from the spectra obtained before renewed addition of SBIZrMe_2_ (Figure 2). This procedure eliminates minor spectral contributions of the binuclear cations (SBIZr)_2_CH_3_)_2_(μ-CH_3_)^+^ with λ_max_ = 429 nm [38], the ^1^H NMR signals of which had reappeared after renewed addition of SBIZrMe_2_ and then remained stable at a level of ca. 8% of the total SBIZr content. 

A slow absorbance decrease between 450 and 500 nm and a concomitant absorbance increase around 600 nm indicate some instability of the cation SBIZr-Me^+^ in the solutions studied. While the identity of any decomposition product(s) was not established, one observes in these toluene-d_8_ solutions of SBIZr-Me^+^ a slow evolution of CH_3_D (Appendix A). Since the only source of deuterium in the reaction system is the deuterated solvent, one decomposition pathway of SBIZr-Me^+^ appears to be its attack at a solvent molecule, probably under formation of Zr-CD_2_C_6_D_5_ or Zr-C_6_D_4_CD_3_ complexes. Despite these complications, we can safely assume that the cation SBIZr-Me^+^ is characterized by the UV-vis absorbance band with λ_max_ = 485 nm shown in Figure 2.

The cation SBIZr-Me^+^ might not be the most suitable model for a SBIZr-polymeryl^+^ cation, however, since the latter is generally assumed to have its polymer chain bound to the Zr center by agostic interactions, possibly to the extent of relegating the B(C_6_F_5_)_4_^−^ counter-anion to an outer-sphere position, while SBIZr-Me^+^ B(C_6_F_5_)_4_^−^ has the anion directly coordinated to its Zr center [34]. As proposed by Bochmann and coworkers, the cation SBIZr-CH_2_SiMe_3_^+^ might be more similar in this regard to a SBIZr-polymeryl^+^ cation, since its Si-Me groups interact with the Zr center strongly enough to prevent coordination of the B(C_6_F_5_)_4_^−^ anion [8]. 

Solutions of SBIZr-CH_2_SiMe_3_^+^ B(C_6_F_5_)_4_^−^ are prepared from SBIZr(Me)-CH_2_SiMe_3_ by an uncomplicated reaction with slightly less than one equivalent of trityl perfluorotetraphenyl borate (to exclude any excess of trityl cation), since formation of binuclear intermediates is not observed in this case. A solution of 1.60 μmol of SBIZr(Me)-CH_2_SiMe_3_ in 1.75 mL of toluene-d_8_ gave, after addition of 0.29 mL of a 4.5 mM solution of trityl perfluorotetraphenyl borate in toluene-d_8_, NMR spectra as expected for the SBIZr-CH_2_SiMe_3_^+^ cation [8], with SiMe_3_ and diastereotopic CH_2_ signals, at −0.73 ppm and at −0.40 and 2.32 ppm, respectively, in a ratio of 9:1:1. The UV-vis spectrum of this reaction mixture, as measured in a 5-mm OD NMR tube, shows a main absorbance band at 468 nm. A shallow absorbance stretching out to 650–700 nm increases—as in the case of SBIZr-Me^+^—slowly with time (Appendix A). 

From the absorbance maxima of SBIZrMe^+^ at 485 nm and of SBIZr-CH_2_SiMe_3_^+^ at 468 nm, we can roughly estimate molar absorptivity values of ca. 1.4·10^3^·M^−1^·cm^−1^ and 2.3·10^3^ M^−1^·cm^−1^, respectively. These estimates are of a similar order of magnitude as molar absorptivity values of 2.5·10^3^ M^−1^·cm^−1^ and of 1.2·10^3^ M^−1^·cm^−1^ observed for the cation SBIZr(μ-Me)_2_AlMe_2_^+^ at 495 nm and for SBIZr-allyl^+^ cations at 560 nm. 

Both, the prototypical SBIZr-alkyl cation SBIZr-Me^+^ as well as its congener SBIZr-CH_2_SiMe_3_^+^ with its rather strong agostic Zr-alkyl interactions, thus have their main absorbance bands at wavelengths between 450 and 500 nm. SBIZr-polymeryl cations—with presumably intermediate degrees of agostic Zr-alkyl interactions—can thus be expected to have their main absorbance band likewise at wavelengths in this region. 

### 3.3. UV-vis Spectra and Concentration Profiles of Species **C-1** and **C-2**

In order to arrive at more quantitative results with regard to the UV-vis spectra of the catalyst species **C-1** and **C-2** and their respective concentration profiles, we have analyzed our UV-vis data (Figure 1) by use of the Multivariate Curve Resolution-Alternating Least Square (MCR-ALS) methodology developed by Tauler and coworkers [39,40]. This tool is designed to decompose data matrices, such as those representing the spectral changes shown in Figure 1, into the spectra and concentration profiles of each of the species involved, while adhering to certain constraints. These constraints might concern—apart from self-evident features such as closure (mass balance) and the non-negativity of absorbance and concentration values—also more specific constraints such as equality or inequality conditions concerning previously determined spectra or concentration values of individual species, which can be used to restrict the range of otherwise possible solutions. At this point we should remark that chemometric analysis procedures such as MCR-ALS will indeed often yield, instead of a straight-forward result, a range of possible solutions. The task to narrow this range of possible solutions to a definitive proposition concerning a specific system, such as ours, will generally require considerations that are rather complex, necessarily making some of the following text somewhat difficult to follow.

Concerning the number of species contributing to the spectral changes considered, we assume, as discussed above, that the absorbance changes shown in Figure 1 are accounted for by interconversion of three spectroscopically distinguishable species, **C-0**, **C-1,** and **C-2**. In accord with this assumption, the resulting spectra and concentration profiles of these three species account for 99.988% of the variance of the experimental data; the residual lack of fit of 1.1% is likely to be below the level of accuracy of our spectroscopic measurements. 

As constraints we use NMR-derived relative concentrations of the pre-catalyst **C-0**, normalized with respect to the total Zr concentration, [C-0]*_t_* = c(C-0)*_t_*/c(Zr)_tot_. For *t* = 0, i.e., before addition of 1-hexene, we set [C-0]*_0_* = 1. For all other times the values of [C-0]*_t_* are set to be greater than 0.03, the minimum value determined by ^1^H NMR. 

The results of such MCR-ALS analyses will depend on the initial estimates of spectra or concentration profiles, from which the algorithm is started on its least-squares minimization of the deviations between calculated and experimental data. For a first approach, MCR-ALS was started from the “purest spectra”, i.e., from UV-vis spectrum vectors with minimal mutual overlap (for details see [40]). In this manner, the UV-vis data shown in Figure 1 yielded for species **C-0**, **C-1** and **C-2** the spectra and concentration profiles labeled as case **a** in Figure 3. Different results, labeled as case **b** in Figure 3, are obtained if the MCR-ALS algorithm is started—with the same constraints for [C-0]*_t_* as before – from an initial estimate of concentration profiles, which is derived from those obtained for case **a** by increasing all values of [C-1]*_t_* by an increment of 1.92·[C-1]*_t_* while decreasing all values of [C-2]*_t_* by the same increment, with all values of [C-0]*_t_* being kept unchanged. This procedure generates a linear combination of the concentration profiles of **C-1** and **C-2**, for which the values of [C-2]*_t_* are close to zero (<0.10) for all *t* < 120 s, i.e., for reaction stage 1.

Cases **a** and **b** represent limits of a range of possible resolutions of the spectral data shown in Figure 1. Case **a** is close to the limit set by non-negative absorbance values (species **C-1** at ca. 630 nm), while case **b** is close to the limit set by non-negative concentration values (species **C-2** at *t* < 120 s). Both resolutions of the experimentally obtained data matrix are equally valid in the sense that the standard deviations (1.09%) between experimentally obtained and calculated absorbance values are the same for both cases. Cases **a** and **b** thus fit the experimentally obtained absorbance data equally well, while fulfilling the applied constraints, a feature known as ‘rotational ambiguity’ [41,42,43,44,45,46].

Both cases reproduce the spectrum of the pre-catalyst SBIZr(μ-CH_3_)_2_Al(Me)_2_^+^ (**C-0**) with its characteristic band at λ_max_ = 495 nm. In both cases practically identical spectra are likewise obtained for species **C-2** (Figure 3A). Their broad absorbance band with λ_max_ = 560 nm is closely similar to that observed for bona-fide SBIZr-allyl cations [28], thus supporting the notion that SBIZr-allyl cations constitute species **C-2**. For the intermediary species **C-1**, however, distinctly different spectra result for cases **a** and **b**.

For case **a**, we obtain for species **C-1** a spectrum with a main band at λ_max_ = 475 nm and with additional bands at 570 and ca. 700 nm. The position of the band at 475 nm is in accord with λ_max_ values between 450 and 500 nm obtained for SBIZr-alkyl cations in Section 3.2 above and would thus suggest that species **C-1** consists of SBIZr-polymeryl cations, i.e., of the species responsible for polymer chain growth in our catalyst system. 

The additional band at ca. 570 nm as well as a shoulder at ca. 495 nm might be computational artefacts caused by an incomplete resolution of the spectrum vector of **C-1** from those of **C-0** and **C-2**. The absorbance band appearing at ca. 700 nm, however, seems to be a proper part of the UV-vis spectrum of species **C-1**. Since no absorbance at such long wavelengths was observed for the SBIZr-alkyl cations considered above, the origin and nature of this absorbance band remain to be clarified.

For case **b**, the spectrum obtained for species **C-1** is quite different from that for case **a**. With an absorbance minimum (instead of a maximum) between 450 and 500 nm and with a broad main band at 570 nm, it is closely similar to the spectrum obtained for species **C-2**, from which it differs only by a shallow shoulder at ca. 700 nm. Prima facie, case **b** would thus suggest that species **C-1**, like **C-2**, consists of Zr-allyl cations.

With regard to concentration profiles (Figure 3B), species **C-0** shows, for case **a** as for case **b**, a rather rapid initial decline and later a partial recovery, as expected from the changes in A(495) shown in Figure 1D. Species **C-1** reaches, in both cases, its maximal concentration after reaction times of ca. 120 s. In case **a**, the maximal value of [C-1]*_120_* is 0.26, while in case **b**, [C-1]*_120_* = 0.75 is about three times higher than in case **a**. [C-1]*_t_* falls below values of 0.1 after ca. 600 s and goes to zero thereafter, in line with the appearance of a new isosbestic point at longer reaction times (cf. Figure 1), which indicates that only species **C-0** and **C-2** remain in the reaction system during reaction stage 3. 

Rotational ambiguities of this kind are to be expected when a reaction intermediate arises with a spectrum overlapping those of the starting and end products, in particular, when its concentration remains comparatively low [41,42,43,44,45,46]. Until this ambiguity is resolved by appropriate means, two mutually incompatible scenarios thus have to be kept under consideration: For case **a**, a main absorbance band of species **C-1** similar to those of the cations SBIZr-Me^+^ and SBIZrCH_2_SiMe_3_^+^ would support the plausible assumption that **C-1** consists of SBIZr-polymeryl cations. This is connected, however, with the non-trivial notion that species of types **C-1** and **C-2** should both arise simultaneously from the pre-catalyst **C-0**. This assumption is compatible with the appearance of the isosbestic point at 532 nm, since during reaction stage 1 species of type **C-1** and **C-2** arise in constant proportion ([C-2]*_t_*_<120_/[C-1]*_t_*_<120_ = 1.9 ± 0.1), as well as with its disappearance during reaction stage 2, where this proportionality is lost.For case **b**, on the other hand, the virtual absence of species **C-2** during stage 1 would be an intuitively attractive explanation for the occurrence of an isosbestic point at 532 nm during that reaction stage. The notion, however, that both the chain-propagating intermediate **C-1** and the final species **C-2** should be of the SBIZr-allyl^+^ type, as implied by their closely similar UV-vis spectra in case **b**, is not in line with generally accepted assumptions concerning catalyst systems of the kind considered here.

Aiming at a resolution of this ambiguity, we have sought to find reaction conditions where intermediate **C-1** arises in our catalyst systems in increased proportions, since the range of acceptable resolutions of the experimental data would then be expected to be diminished, possibly to the point of excluding one of the two scenarios described above.

### 3.4. Catalyst Re-Activation

In a recent study, it had been shown that SBIZr(μ-Me)_2_Al(Me)_2_^+^-based catalyst systems, which are largely converted to Zr-π-allyl species after polymerizing an excess of 1-hexene, will polymerize a second portion of monomer at a similar rate as the first portion [28], presumably by re-converting most or all of species **C-2** to chain-carrying intermediates of type **C-1**. We have now tried to quantify this monomer-induced change in catalyst speciation.

Spectral changes produced by addition of a second portion of 1-hexene to such a pre-used catalyst solution are less spectacular than those seen after addition of the first portion. Clearly discernible absorbance increases are observed, however, at ca. 470 and 700 nm, i.e., in regions characteristic for species **C-1**, while the absorbance decreases at ca. 560 nm, in a region characteristic for species **C-2**.

Analysis of these rather small spectral changes by MCR-ALS methods, using as constraints the Spectra obtained for each of these species after the first addition of 1-hexene (cf. Figure 3A), leads – for case **a** as for case **b** – to concentration profiles for species **C-1** which rise to a level comparable to that previously observed upon addition of the first portion of 1-hexene, i.e., to [C-1]*_t_* = 0.3 for case **a** and to [C-1]*_t_* = 0.55 for case **b**. 

While increased levels of [C-1]*_t_* were thus not achieved in this way, the rate at which species **C-1** re-appears upon addition of a second portion of 1-hexene is much higher than that observed after addition of the first portion. Whereas the pre-catalyst **C-0** is transformed to species **C-1** with a half-live of *t*_1/2_ = 50 s after a first addition of 1-hexene, re-conversion of species **C-2** to **C-1** appears to be completed already when a first spectrum is taken at *t* = 6 s, thus setting an estimate of *t*_1/2_ ≤ 3 s for this process. Utilization of this observation for a low-temperature experiment is to be described in the next section of this manuscript. 

### 3.5. Temperature Effects

In trying to find reaction conditions that would afford more of species **C-1** and less of **C-2**, such that the spectral features of **C-1** would become more clearly discernible, catalytic polymerization reactions were run at different temperatures. Results of reactions run a 40 °C and 50 °C and their analysis by MCR-ALS methods did not afford any confinement of the ambiguity limits. On the contrary, the ‘ambiguity gap’ between the limiting cases **a** and **b** increased: at 50 °C maximal normalized concentrations values of 0.08 and 0.90 were obtained for species **C-1** for case **a** and case **b**, respectively, as compared with much closer respective values of 0.26 and 0.75 obtained at 23 °C.

This result would indicate that ambiguity gaps should become narrower at lower temperatures. A catalytic reaction run at −20 °C did not give useful results, however. At this temperature, the absorbance of species **C-0** decreased only very slowly, over hours instead of minutes, with a final value of [C-0]*_t_* estimated at ca. 60%, instead of ca. 3% as found for reactions run at 23 °C. Apparently, release of SBIZr-Me^+^ from the pre-catalyst SBIZr(μ-Me)_2_AlMe_2_^+^ becomes very slow at lower temperatures. As a consequence, large residual concentrations of **C-0** overshadow the spectral features of species **C-1** to such an extent that neither [C-1]*_t_* values nor the spectrum of **C-1** can be reliably assessed.

In order to observe a more extensive formation of species **C-1** in the course of a low-temperature polymerization reaction, we have conducted a catalyst re-activation experiment at −30 °C, thereby exploiting the much higher rate, with which species **C-1** is generated from species **C-2** than from **C-0**. For this experiment, a first portion of 0.25 mL of 1-hexene was added in the glovebox to 2.3 mL of the standard catalyst solution contained in a 10-mm UV-vis cuvette. After a reaction time of 20 min, the violet-colored reaction mixture was cooled, still in the glovebox, to −30 °C by inserting the cuvette into a rectangular well in a pre-cooled aluminum metal block and then closed with a PTFE-lined septum cap. This procedure precludes that upon cooling the argon pressure inside the cuvette sinks below the external air pressure and, hence, that air could enter when the cuvette closure is punctuated for the second addition of 1-hexene. After transferring the cooled cell to a spectrometer cell holder pre-cooled to −30 °C, a second portion of 0.25 mL of 1-hexene was injected through the septum closure and the spectral changes ensuing at that temperature were recorded as usual. 

The spectral traces thus obtained during the first 300 s (Figure 4A) show a substantial absorbance increase between 450 and 500 nm, an even greater decrease around 560 nm and a small but significant absorbance increase around 700 nm. These changes indicate that substantial fractions of species **C-2** are converted to species **C-1** upon addition of the second portion of 1-hexene, while the observation of an isosbestic point at 505 nm indicates—in accord with our findings at −20 °C—that the residual concentration of the pre-catalyst, [C-0]*_t_* with λ_max_ = 495 nm, is hardly changing at all. At later reaction times, the transformation of **C-2** to **C-1** is reversed at a very slow rate (Figure 4B).

Evaluation of the experimental data thus obtained by MCR-ALS methods (Figure 5) indicates substantial diminution of the rotational ambiguity compared to that associated with the spectral data obtained at 23 °C. The spectra obtained at −30 °C for species **C-1** at the limit of non-negative absorbance values, now labeled as case **a’**, and at the limit of non-negative concentration values (case **b’**) do not enclose, in particular, the spectrum obtained at 23 °C for species **C-1** at the limit of non-negative concentration values (case **b**), such that the latter does not appear to be a valid solution for our reaction system.

In order to assess the limits of acceptable solutions, we assume—based on the qualitative observations described above—that our catalyst system comprises the same three species, **C-0**, **C-1** and **C-2**, at −30 °C as at 23 °C, and that the spectrum of each species at −30 °C does not differ greatly from that at 23 °C. For symmetry-allowed charge-transfer absorbance bands such as those considered here, much less temperature dependence is to be expected than for symmetry-forbidden d-d* transitions, which owe their (rather low) intensity to vibronic coupling with charge-transfer transitions at higher energies. The UV-vis absorbance band of the pre-catalyst cation SBIZr(μ-Me)_2_AlMe_2_^+^, for example, remains practically unchanged between −20 °C and +40 °C (Appendix A). With these assumptions, the range of acceptable spectra for each species can be determined by performing an MCR-ALS analysis on a combined data set [40], i.e., on the absorbance values obtained at 23 °C stacked upon those obtained at −30 °C (Figure 6).

Such a combined analysis yields spectra within a range quite similar to that obtained from the re-activation reaction at −30 °C alone (Figure 6A), in accord with the notion that this data set is limiting the range of acceptable solutions. For species **C-1**, in particular, all acceptable spectra have an absorbance maximum between 450 and 500 nm and would thus rule out the scenario of case **b**, which yields an absorbance minimum in this range. 

Application of the spectra shown in Figure 5A as constraints for an MCR-ALS analysis of the data set obtained at 23 °C now gives concentration profiles for species C-1 and C-2 with relatively narrow ambiguity zones (Figure 6B). From these, we can assess ‘mean’ concentration values for [C-1]*_t_* and [C-2]*_t_*. Cases **a’** and **b’** then set an uncertainty limit of ca. ±30% for these ‘mean’ concentration values of catalyst species C-1. Despite their limited accuracy, these concentration profiles clearly rule out the consecutive formation of species C-1 and C-2, previously associated with case **b**. Instead, they strongly support the first of the two scenarios discussed above—i.e., the simultaneous formation of **C-1** and **C-2** during reaction stage 1—as the only valid description of our catalyst system. 

### 3.6. Reaction Rates and Mechanisms

The interconversion reactions between catalyst species **C-0**, **C-1** and **C-2** discussed above are initiated and driven by the presence of 1-hexene in the catalyst system. Therefore, concentrations of these species and their time-course must be connected with the decaying concentrations of the olefin, [hex]*_t_* = c(1-hexene)*_t_*/c(1-hexene)_0_, and vice versa. In order to address these interconnections, we have determined, from the intensities of the ^1^H NMR signals of 1-hexene at 4.98 and 5.75 ppm and of poly-1-hexene at 1.70 ppm, how the values of [hex]*_t_* decrease with time (Figure 7).

A similar decline as for [hex]t holds also for the rate of consumption of 1-hexene, −d[hex]t/dt, in line with the assumption that this rate should decrease together with [hex]t. The quantity (−d[hex]t/dt)/[hex]t, however, i.e., the rate of consumption of 1-hexene at each time t divided by the residual concentration of 1-hexene at that time, shows a time profile which parallels that of species C-1 over much of the reaction time (Figure 7). For the ratio of these two variables, (−d[hex]t/dt)/([hex]t·[C-1]t), we obtain a value of 0.071 ± 0.018 s^−1^ for t < 600 s, i.e., for stages 1 and 2.

This observation implies that the rate of 1-hexene polymerization by the present catalyst system, −dc(hex)*_t_*/dt = k_P_·c(C-1)*_t_*·c(hex)*_t_*, is of second order, first-order with regard to the concentration of **C-1** as well as to that of 1-hexene, The rate-determining step of the catalytic polymerization thus involves attack of 1-hexene at zirconocene complexes of type **C-1**, which must thus contain a Zr-polymer bond. The data presented above, together with a total catalyst concentration of c(Zr)_tot_ = 0.5 mM, yield for this polymer-growth step an apparent second-order rate constant of *k*_P_ = 130 ± 35 M^−1^·s^−1^. This estimate agrees with rate constants of *k*_P_ ≥ 100 M^−1^·s^−1^, determined by Landis and Christianson for SBIZr-catalyzed 1-hexene polymerizations by stopped-flow experiments at room temperature [26].

These catalyst systems produce, under conditions close to those described above, polyhexene samples with molar mass values of M_W_ = 41,600 ± 1500 and M_N_ = 24,000 ± 1000 (PDI = 1.75), as measured by GPC analysis. This corresponds to a number-averaged degree of polymerization of P_N_ = 286 ± 12. With a ratio of c(1-hexene)0/c(Zr)0 = 640 used in these catalyst systems, each Zr center present would thus on average produce 2-3 polymer chains during each experiment. Zr-alkyl and Zr-polymeryl species of type **C-1** make up only 20–25% of the total Zr-content, however (Figure 6). About 10 chains will thus on average grow on each cation of type **C-1** within several minutes. 

For a more detailed analysis of 1-hexene consumption and the accompanying chain growth, we have to consider the concrete chemical nature of the zirconocene complexes subsumed under each species type (Scheme 1). While species **C-0** simply represents the cation SBIZr(μ-Me)_2_AlMe_2_^+^ (**1**), species **C-1** comprises a multitude of ion pairs containing different SBIZr-alkyl^+^ and SBIZr-polymeryl^+^ cations. Of these, we have collected in Scheme 1 some of the most obvious conceivable structures (**2**-**8**) and the reaction paths which lead to—and further onward from—each of them. Complexes with 1,2-inserted chain ends, such as **3, 4** and **6**, as well as their counterparts with internally unsaturated polymer chains (**7** and **8**) are likely to insert an olefin with a rather similar rate constant, k(ins-1). The cation SBIZr-Me^+^ (**2**), on the other hand, is known to insert an olefin with a substantially smaller rate constant (k(ins-0)) than structures **3**-**8** [5]. The same appears to hold for regio-irregular 2,1-insertions leading to structure **5**, (k(ins-2)), as well as for regio-regular insertions into 2,1-inserted structures such as **5** [7]. The value of *k*_P_ = 130 ± 35 M^−1^·s^−1^, obtained above from our experimental data must thus be some weighted average of multiple insertion rate constants, but might be rather close to k(ins-1), since regio-regular insertions into primary SBIZr-polymeryl bonds probably constitute the major reaction path for 1-hexene consumption.

Of the reactions which convert species **C-0**, **C-1,** and **C-2** to each other in the course of the catalytic process, we first consider the reaction by which pre-catalyst **C-0** is consumed upon addition of 1-hexene. A strictly linear plot of ln[C-0]*_t_ versus t* (Appendix A) shows that during reaction stage 1 (*t* < 120 s) the rate of consumption of **C-0**, −d([C-0]*_t_*)/d*t*, is independent of the concentration of 1-hexene and first-order with respect to [C-0]*_t_*. 

We can thus assume that dissociation of SBIZr(μ-Me)_2_AlMe_2_^+^ to SBIZr-Me^+^ and AlMe_3_ is rate-determining and that SBIZr-Me^+^ thus released will react completely with the large excess of 1-hexene present during reaction stage 1. For *t* < 120 s, the reaction rate of **C-0** will thus be equal to the rate of its dissociation, −d([C-0]*_t_*)/d*t* = k(diss)·[C-0]*_t_*, with a dissociation rate constant of k(diss) = 0.015 s^−1^. At longer reaction times, when the concentration of 1-hexene becomes too low to out-run the re-association of SBIZr-Me^+^ with Al_2_Me_6_, the rate of consumption of **C-0** is approximated by the relation d([C-0]*_t_*)/d*t* = k(diss)·[C-0]*_t_*·[hex]*_t_*/([hex]*_t_*+k(reass)), with k(reass) = 0.075 M. 

From the – now reliably confined – concentration profiles shown in Figure 6B it appears certain that the reaction of SBIZr(μ-Me)_2_AlMe_2_^+^ (**C-0**) with 1-hexene gives rise to species of type **C-1** as well as of type **C-2**, i.e., to SBIZr-alkyl^+^ as well as to SBIZr-allyl^+^ complexes. Apparently, 1-hexene can react with the cation SBIZr-Me+ not only by insertion into the Zr-Me bond, but also under σ-bond metathesis [47,48], i.e., by transfer of one of its allylic H atoms to the Zr-bound Me group, which is thus released as methane (Scheme 1).

The concentration profiles shown in Figure 6 indicate that SBIZr-alkyl^+^ complexes of type C-1, such as **4** or **5**, and SBIZr-allyl^+^ complexes, such as **3**, arise initially in parallel in comparable fractions from the reaction of **C-0** with 1-hexene. After about 120 s, [C-1]*_t_* begins to decline under formation of species of type **C-2**. For the time period between 300 s (where [C-0]*_t_* is substantially level) and 600 s, a monomolecular decay of species of type **C-1** to a Zr-allyl species **C-2** such as complex **9**, with a rate constant of k(de-ac) = 4.6 × 10^−3^ s^−1^ is indicated by a plot of ln[C-1]*_t_*against *t* (Appendix A). Monomolecular decay of Zr-alkyl cations such as **5** to Zr-allyl cations such as **9** probably occurs under loss of H_2_ (via an intermediary Zr-hydride-olefin complex produced by ß-H transfer to the metal center), a reaction route discovered already early on [12], and described as a particularly facile process in a more recent Density-Functional Theory study [17].

All SBIZr-polymeryl^+^ complexes of type **C-1**, which reach their maximum of ca. 20–30% of the total catalyst content around *t* = 120 s, appear to end up as polymer-carrying SBIZr-allyl complexes such as **9**. At the end of each experiment, SBIZr-allyl complexes of this type will thus likewise make up ca. 20–30% of the catalyst content, while the initially formed SBIZr-allyl complexes **3** will then still constitute ca. 70–80% of the catalyst content. Species **C-2** will then most likely consist of complexes **3** and **9** in a ratio of ca. 3:1. 

The re-conversion of species **C-2** to pre-catalyst **C-0** by methyl-*vs*.-allyl exchange between species **C-2** and Al_2_Me_6_, which slowly proceeds during stage 3, is approximated by an apparent first-order reaction, d[C-0]*_t_*/d*t* = −d[C-2]*_t_*/d*t* = k(exch-9)·[C-2]_t_, with an apparent rate constant of k(exch-9) = 6.3 × 10^−5^ s^−1^. Rate and extent of the re-conversion of species C-2 to pre-catalyst C-0 by methyl-vs.-allyl exchange are found to increase with the concentration of Al_2_Me_6_ in the catalyst reaction system, i.e., with the initial [Al]/[Zr] ratio (Appendix A). Since part of the added Al_2_Me6 is likely to be consumed as a scavenger by trace impurities, our data are not sufficient to evaluate the rate law of this exchange reaction more accurately.

Further relevant rate data can be obtained from the catalyst re-activation experiments described in Section 3.4 of this study. An MCR-ALS analysis of the (rather small) spectral changes produced by addition of a second portion of 1-hexene, now by the use of spectra for species **C-0**, **C-1** and **C-2** derived from the catalyst re-activation experiment at −30 °C as constraints, yields the concentration profiles shown in Figure 8. Most striking in Figure 8 is a rapid re-activation of species **C-2** to **C-1**, which occurs within 6 s after addition of 1-hexene and then stops abruptly at a level similar to the level reached by [C-1]*_t_* after the first addition of 1-hexene. A plausible explanation for this coincidence appears to be the assumption that 1-hexene can re-activate only that part of **C-2** which is made up by Zr-allyl complexes of type **9**, i.e., the successor(s) of all or most of the Zr-polyhexyl complexes of type **C-1** formed during the first polymerization run. For this reaction, we would assume a second order rate law, −d[9]*_t_*/d*t* = k(re-ac)·[9]*_t_*·[hex]*_t_* with a rather high rate constant k(re-ac) in the range of ca. 0.1–0.5 M^−1^·s^−1^. Although high in comparison to the rates of other interconversion reactions between catalyst species considered here, catalyst re-activation by olefin insertion into Zr-allyl bonds of complexes of type **9** would thus still be lower by 2 to 3 orders of magnitude than that of hexene insertion into Zr-polymeryl bonds of complexes of type **4** or **5**.

The initially formed Zr-allyl complex **3**, on the other hand, would thus appear to be quite inert against re-activation by 1-hexene on the time scale of our experiments. At first sight, it might appear counter-intuitive, that a complex of type **9** with its more bulky allyl ligand should be more reactive than the much less crowded complex **3**. Our main point here is, however, that a stable Zr-η^3^-allyl unit is presumably not amenable to insertion of an olefin at all. Only to the degree that it is converted to its Zr-η^1^-allyl isomer will such a unit likely become accessible to be re-activated by olefin insertion. That steric repulsion between substituents at the allyl ligand and at the aromatic ligand framework favor the formation of η^1^-haptomers in equilibrium with zirconocene-π-allyl cations has been shown in an elegant ^1^H NMR study by Vatamanu [24]. Complex **9** would thus be much more likely than complex **3** to be transformed to its η^1^ isomer. Its Zr-bound η^1^-allylic chain end would then have very similar steric demands as the normal Zr-bound 1,2-inserted chain end in complex **5** and might thus approach the insertion rate of the latter, while the sterically unencumbered Zr-η^3^-allyl complex **3** remains relatively inert.

## 4. Discussion

Concurrent formation of SBIZr-σ-polyhexyl and SBIZr-π-allyl cations is indicated by an analysis of UV-vis spectral changes associated with the addition of excess 1-hexene to solutions of the pre-catalyst SBIZr(μ-Me)_2_AlMe_2_^+^ (F_5_C_6_)_4_B^−^ at 23 °C, in particular by elimination of alternative resolutions of the raw UV-vis spectroscopic data thus obtained through use of UV-vis data derived from a catalyst re-activation experiment conducted at −30 °C. The SBIZr-allyl species formed during the initiation phase are proposed to arise by a σ-bond metathesis reaction between 1-hexene and SBIZr-Me^+^. Two types of SBIZr-π-allyl^+^ species are thus proposed to occur in these catalyst systems:cations of composition SBIZr-η^3^-(1-R-C_3_H_4_)^+^ (R = n-propyl) arising during the initiation reaction between SBIZr-Me^+^ and 1-hexene by σ-bond metathesis under release of methane,polymer-containing cations SBIZr-η^3^-(1-R-2-pol-C_3_H_3_)^+^ (pol = *i*-polyhexyl) formed from SBIZr-σ-polyhexyl^+^ cations, e.g., by loss of H_2_, during later reaction stages.

While appearance of the latter kind of η^3^-allyl species during later stages of zirconocene-catalyzed olefin polymerizations is well-documented in the literature [10,26,28], initial formation of SBIZr-η^3^-(1-n-propyl-C_3_H_4_)^+^ cations has not been reported so far. Such a reaction is not altogether unlikely, however.
Zirconocene-η^3^-allyl^+^ complexes are known to arise from zirconocene-Me^+^ cations by a σ-bond metathesis reaction with 1,1-disubstituted olefins, for which the insertion pathway is inaccessible [15,23].Olefin insertions into a zirconocene-Me^+^ bond are known to be much slower than insertions into a zirconocene-polymeryl^+^ bond [5]; σ-bond metathesis might thus be more competitive here.The formation of CH_3_D from SBIZr-Me^+^ and toluene-d_8_, as described in Section 3.2. above, indicates a pronounced tendency of SBIZr-Me^+^ cations toward σ-bond metathesis.

The proposal that two types of SBIZr-η^3^-allyl cations occur in these catalyst systems is supported by the observation that a second addition of excess 1-hexene converts back to catalytically active SBIZr-σ-polyhexyl cations only that part of SBIZr-π-allyl cations, which corresponds to the fraction of SBIZr-σ-polyhexyl cations formed in the first reaction run and hence now present as cations of type SBIZr-η^3^-(1-R-2-pol-C_3_H_3_)^+^, while leaving the fraction of the initially formed cations of type SBIZr-η^3^-(1-R-C_3_H_4_)^+^ unchanged.

Nevertheless, this proposal should be considered as tentative as long as the initial formation of cations such as SBIZr-η^3^-(1-n-propyl-C_3_H_4_)^+^ is not established more directly, e.g., by NMR methods. Attempts to differentiate between the UV-vis spectra of ‘early’ SBIZr-η^3^-(1-R-C_3_H_4_)^+^ and ‘late’ SBIZr-η^3^-(x-R-(3-x)-pol-C_3_H_3_)^+^ cations (R = n-propyl, pol = *i*-polyhexenyl, x = 1 or 2), e.g., by a four-component MCR-ALS analysis or by division of the data set into an ‘early’ and a ‘late’ set, did not yield meaningful results. If found to be true, however, this proposal would have a number of interesting corollaries.
The occurrence of Zr-allyl species, which cannot be re-activated by monomer, might contribute to observations that only a fraction of the total zirconocene content of such polymerization catalyst systems seem to be actively involved in chain growth, while major portions remain in some ‘dormant’ state. Based on analyses of a series of polypropylene samples, Busico, Cipullo and coworkers have proposed that the deactivation of propagating centers is caused by 2,1-misinsertions [7]. To which degrees such 2,1-misinsertions and/or the formation of Zr-allyl species contribute to the observed deactivation might be clarified by further studies on the microstructures of the polymer products and by suitable quenching experiments.The presence of alkyl or polymeryl substituents at the central position of polymer-carrying Zr-allyl cations, such as complex **9**, will undoubtedly cause substantial steric strain with the ligand framework of these SBIZr complexes. Even greater steric strain is to be expected for homologous complexes with extended aromatic ring ligands above and below the equatorial ligand plane, such as dimethylsilyl-bridged bis-benzindenyl or bis-4-phenyl-indenyl zirconium derivatives [49,50]. Catalysts with such extended ligands might thus owe their particularly high activities to a destabilization of polymer-carrying Zr-allyl cations of type **9**, i.e., to an increased rate of their re-conversion to Zr-polymeryl species and/or to a suppression of their formation.The occurrence of polymer-carrying Zr-allyl cations such as **9**, which are easily re-converted to Zr-polymeryl cations by relatively fast monomer insertion, might explain an apparently higher than first-order dependence of polymerization rates on monomer concentrations, repeatedly observed with various zirconocene-based polymerization catalyst systems. As shown by Fait et al. [18], this phenomenon is best explained by changes from an inactive to an active catalyst state induced by higher monomer concentrations. Such a situation would be expected to pertain for zirconocene-catalyzed olefin polymerizations run within certain concentration and temperature regimes, due to a monomer-dependent interconversion between catalytically active Zr-polymeryl and polymer-carrying Zr-allyl cations described above.For practical use, zirconocene-based olefin-polymerization catalysts have to be immobilized on some solid support, usually on a suitable silica gel. Frequently, these supported catalyst are subjected, before actual use, to a ‘pre-polymerization’ procedure, i.e., to a treatment with some olefin monomer, in order to package each catalyst grain in a thin polymer layer, which stabilizes it against crumbling, but allows its shrapnel-like expansion into minute fragments during the proper polymerization run [51,52]. In view of our study, such pre-polymerized zirconocene catalyst are likely to be present on their supports in form of their Zr-π-allyl derivatives and should hence follow re-activation kinetics similar to that shown in Figure 8 above, rather than the much slower activation kinetics of a catalyst precursor of type Zr(μ-Me)_2_AlMe_2_^+^.

To test these hypotheses, further studies concerning the formation and re-activation of Zr-allyl cations in catalyst systems with differing ligand structures would be called for. Similar studies with other monomers, such as propene and ethene and their mixtures, might reveal how the occurrence of Zr-allyl cations also influences catalyst systems practically used, e.g., for the production of linear low-density olefin copolymers.

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
