# Peer review of "Catalyst Speciation during ansa-Zirconocene-Catalyzed Polymerization of 1-Hexene Studied by UV-vis Spectroscopy—Formation and Partial Re-Activation of Zr-Allyl Intermediates"

_polymers, 2019, doi:10.3390/polym11060936_

Reviewer 1 Report

Manuscript ID polymers-503797

This is a great study which contains several interesting findings and would be of high interest to the community. There are only a few minor comments/suggestions:

1) In the literature, there is a certain “dualism” in identifying the possible deactivation mechanisms for ZN-type and related catalytic systems. In this study as well as in previous contributions, the Authors systematically observed the formation of dormant allyl species reluctant to further insertions of alpha-olefins.  At the same time, the group of Busico/Cipullo by having analyzed a series of polypropylene samples suggested that 2,1-misinsertions are responsible for the deactivation of propagating centers (ref. [7] + ACIE 2016, 55, 8590). What if both reactions could operate for the same catalytic system and be the key steps of a global multi-step deactivation mechanism ?

Within the reactivity pattern proposed on Scheme 1, the activity of a bulkier species 10 and complete reluctance of the nearly isostructural species 9 towards insertion of hexene-1 are confusing and not understandable. Whether the actual structure of the allyl complex 9 could be somewhat different than that proposed by the Authors ? For instance, thanks to a known deficient regioselectivity of the non-substituted (SBI) catalyst the very first monomer insertion can be once in a while regioirregular (2,1-) resulting in a secondary alkyl product (SBI)Zr-CH(Et)CH2CH2CH2CH3. Then, the latter evolves into the one of the possible corresponding allylic products, (SBI)Zr…(Me-CH-CH=CH-nPr) or (SBI)Zr…(Et-CH-CH=CH-Et), via the mechanism exemplified in lines 647-648. These new much bulkier 1,1’-R2-allylic products should stay unreactive towards further attack of monomer.

2) NMR spectroscopic study conducted for the low molecular weight PHE samples could potentially bring additional clues on the mechanism of reactivation of 10 towards 7. In this case, the resulted polymer should incorporate internal vinylidene groups Pol1-CH2-C=(HR)-CH2-Pol2 potentially distinguishable by NMR spectroscopy from the terminal vinylidene chain-end groups (ref. [14]).  

3) Line 617, in the allylic ligand of 10: there should be CH2 instead of CH3

4) Line 716: [23-25] instead of [10]

Author Response

We agree with reviewer 1 on all issues raised under point 1. We have tried to take care of his/her concerns by extensive changes in the paragraphs following after line 740 and after line 794 of the revised version of our manuscript.

We find it an interesting possibility that 2,1-misinserted chain ends might be converted to Zr-allyl complexes which are substituted at both terminal positions and hence unlikely to be re-activated by insertion of an olefin. Yet, 2,1-misinsertions have been deemed to be rather rare events by Busico, Cipullo and coworkers, whereas formation of Zr-allyl-species during catalyst initiation appears to be about equally frequent as normal 1,2-insertion into the Zr-Me+  pre-catalyst. Without further – e.g. NMR – data to check this issue, we would thus rather not enter into further discussions of this possibility at this point.

            The suggestion made in point 2, that internal C=C double bonds arising from re-activation of Zr-allyl intermediates should be detectable by NMR techniques, is certainly a valid one. This task would exceed the frame of the present UV-vis study, however. Such a study would probably best be done by a research group with special expertise in this methodology, such as the one at the University of Naples.

            Point 3 has been taken care of in the revised version of Scheme 1.

            Point 4 has been corrected in the revised text that is now following after line 740.

            We thank this reviewer for especially detailed and helpful comments.

Reviewer 2 Report

This paper from Brintzinger and coworkers studied the ansa-Zirconocene-catalyzed 1-hexene polymerization with UV-vis spectroscopy formation and partial re-activation of Zr-Allyl intermediates, and studied the accompanying UV-vis spectral changes in more detail. The results indicated that the onset of polymerization catalysis is associated with the concurrent formation of two distinct zirconocene species. It is a rather interesting work. I do not see anything scientifically wrong with the manuscript. So, I suggest to further consideration for publication.

Author Response

Reviewer 2.

            While we appreciate the reviewer’s suggestion for improving our manuscript, we doubt that the difficulty lies with our use of the English language, but rather with the inherent complexities associated with describing data interpretation by the MCR-ALS procedures.  We have tried to explain this circumstance by some remarks at the end of the first paragraph that deals with this difficulty, on lines 285-290. 

            We appreciate that he/she finds our study interesting and scientifically correct.

Reviewer 3 Report

This manuscript describes study on 1-hexene polymerization using zirconocene catalyst monitored by UV-vis spectra.  Although the content of this manuscript should be very interesting, the manuscript is difficult to follow.  It is thus better that the authors consider appropriate revisions for better understanding with readers.  Since the content would introduce interesting fact, I believe that the manuscript can be acceptable after appropriate revisions.  Several suggestions, for example, are as follows.

Figure 1 (caption) and discussion in page 3: More explanation should be necessary.  For example, placing interval of scans in Figure 1A (and Figures 1B and 1C) and what stages 2 and 3 means in the caption (not only description in the text), and consider insertion of expanded spectra in Figures 1B and 1C.  Insertion of scheme would be helpful.

Consider placing expand spectra in Figure 2, and placing additional explanations in case a (a’) and case b (b’) in the figure captions.

Author Response

          Following the suggestions of reviewer 3, we have expanded the legends of Figures 1 to 5, so as to explain in more detail the spectral changes shown, including the time intervals between scans in Figures 1A-1C. In the legend to Figure 1, we have also clarified the meaning of reaction stages 1, 2 and 3. In the legends of later Figures, the meaning of cases a and b, as well as of a’ and b’ is now explained. We have considered adding expanded inserts into some of the Figures, but are of the opinion that this does not result in added clarity.

We have simplified the reaction scheme (Scheme 1) and tried to make it more explicit. Further explanations have also been added in the legend of Scheme 1 and in the paragraph describing the proposed reaction mechanisms, on lines 740-752.

In addition, we have added, following line 285 of the revised manuscript, some remarks to explain the necessarily somewhat complex nature of the MCR-ALS resolution method used.

          We appreciate that through the suggestions of this reviewer, the article has gained in clarity.

Round  2

Reviewer 3 Report

I have read the revised version.  The authors revised the manuscript according to the comments, and the reviewer feels that the manuscript can be accepted.  It may be better to check misleading before opening on the web.